# Non-Achievement of Alanine Aminotransferase Normalization Associated with the Risk of Hepatocellular Carcinoma during Nucleos(t)ide Analogue Therapies: A Multicenter Retrospective Study

**DOI:** 10.3390/jcm11092354

**Published:** 2022-04-22

**Authors:** Jun Inoue, Tomoo Kobayashi, Takehiro Akahane, Osamu Kimura, Kosuke Sato, Masashi Ninomiya, Tomoaki Iwata, Satoshi Takai, Norihiro Kisara, Toshihiro Sato, Futoshi Nagasaki, Masahito Miura, Takuya Nakamura, Teruyuki Umetsu, Akitoshi Sano, Mio Tsuruoka, Masazumi Onuki, Hirofumi Niitsuma, Atsushi Masamune

**Affiliations:** 1Division of Gastroenterology, Tohoku University Graduate School of Medicine, Sendai 980-8574, Japan; ksato9139@gmail.com (K.S.); m-nino5@hello.odn.ne.jp (M.N.); tomoaki0079@gmail.com (T.I.); akitoshi.s.1129@gmail.com (A.S.); ytrsb237@gmail.com (M.T.); m-onuki.1128@outlook.jp (M.O.); hbvdna@kjc.biglobe.ne.jp (H.N.); amasamune@med.tohoku.ac.jp (A.M.); 2Department of Hepatology, Tohoku Rosai Hospital, Sendai 981-8563, Japan; tkobayashi@tohokuh.johas.go.jp; 3Department of Gastroenterology, Japanese Red Cross Ishinomaki Hospital, Ishinomaki 986-8522, Japan; akahane-ttyh@mva.biglobe.ne.jp; 4Department of Gastroenterology, South Miyagi Medical Center, Ogawara 989-1253, Japan; osamu.kim@gmail.com; 5Department of Gastroenterology, Iwaki City Medical Center, Iwaki 973-8555, Japan; takai59829@gmail.com; 6Department of Gastroenterology, Japan Community Health Care Organization Sendai South Hospital, Sendai 981-1103, Japan; kisara-norihiro@sendaiminami.jcho.go.jp; 7LC Clinic, Sendai 980-0021, Japan; t-sato@lc-sendai.jp; 8Department of Gastroenterology, Sendai City Hospital, Sendai 982-8502, Japan; nagafuto@hotmail.com; 9Department of Gastroenterology, Omagari Kousei Medical Center, Daisen 014-0027, Japan; miuramas@nifty.com; 10Department of Gastroenterology, Yamagata City Hospital Saiseikan, Yamagata 990-8533, Japan; ikuyokuruyo888@gmail.com; 11Department of Internal Medicine, Kesennuma City Hospital, Kesennuma 988-0181, Japan; teruo_0923@yahoo.co.jp

**Keywords:** HBV, HCC, NA, on-treatment response, aMAP risk score

## Abstract

Patients with a chronic hepatitis B virus (HBV) infection who are treated with nucleos(t)ide analogues (NAs) are still at risk for hepatocellular carcinoma (HCC), and it has been clinically questioned whether patients with a high risk of HCC can be identified efficiently. We aimed to clarify the risk factors associated with the development of HCC during NA therapies. A total of 611 chronically HBV-infected patients without a history of HCC, who were treated with NAs for more than 6 months (median 72 months), from 2000 to 2021, were included from 16 hospitals in the Tohoku district in Japan. Incidences of HCC occurrence were analyzed with clinical factors, including on-treatment responses. Alanine aminotransferase (ALT) normalization, based on the criteria of three guidelines, was analyzed with other parameters, including the age–male–ALBI–platelets (aMAP) risk score. During the observation period, 48 patients developed HCC, and the cumulative HCC incidence was 10.6% at 10 years. Non-achievement of ALT normalization at 1 year of therapy was mostly associated with HCC development when ALT ≤ 30 U/L was used as the cut-off (cumulative incidence, 19.9% vs. 5.3% at 10 years, *p* < 0.001). The effectiveness of the aMAP risk score at the start of treatment was validated in this cohort. A combination of an aMAP risk score ≥ 50 and non-achievement of ALT normalization could stratify the risk of HCC significantly, and notably, there was no HCC development in 103 patients without these 2 factors. In conclusion, non-achievement of ALT normalization (≤30 U/L) at 1 year might be useful in predicting HCC during NA therapies and, in combination with the aMAP risk score, could stratify the risk more precisely.

## 1. Introduction

Hepatitis B virus (HBV) infection is a major health problem worldwide, and the World Health Organization (WHO) estimates that 296 million people have a chronic HBV infection [1]. They are at risk of liver cirrhosis and hepatocellular carcinoma (HCC), but antiviral therapies can reduce such life-threatening diseases [2,3,4]. Nucleos(t)ide analogues (NAs) effectively suppress HBV replication and are widely used because they can be administered orally, and usually safely, for a long time [5]. Although NAs inhibit the reverse transcription of the HBV genome and rapidly suppress HBV DNA in the serum during therapy, the effect on the reduction in hepatitis B surface antigens (HBsAg) is limited [6]. When viral suppression is not optimal, hepatitis relapses frequently after the cessation of NAs [7]. The reason for the difficulty of HBV eradication is that HBV forms covalently closed circular DNA (cccDNA) in the nucleus after viral entry into hepatocytes [8,9]. HCC rarely occurs when HBsAg is cleared [10], but the rate of HBsAg loss by NAs is low and on-treatment patients still have a risk for HCC [11,12]. Therefore, it has been clinically questioned whether patients with a high HCC risk can be efficiently identified.

Thus far, several risk factors associated with HCC development during NA treatments have been reported [13], and risk scores have been developed [14,15]. Recently, an age–male–ALBI–platelets (aMAP) risk score that estimates HCC risk universally, including patients with HBV, hepatitis C virus (HCV) and non-viral hepatitis, was developed [16]. This score involves only age, sex, albumin (Alb), total bilirubin (T-Bil) and platelet counts (PLT). Basically, old age, male gender and advanced fibrosis of the liver are associated with a high risk of HCC [13]; however, as a feature of HBV-associated HCC, it can sometimes be found in the liver without advanced fibrosis [17]. Pathogeneses such as an integration of the HBV genome into hepatocytes [18] and the accumulation of mutant HBV proteins [19] have been proposed, but they are hard to evaluate in clinical settings. In some recent studies, normalization of alanine aminotransferase (ALT) during NA therapy was reported as being associated with a low risk of HCC [20,21,22], but the cut-off of ALT normalization and the timing of judgment has not been sufficiently evaluated. In this study, we aimed to find clinically available risk factors, including responses to therapies that can be widely obtained by most physicians in a multicenter retrospective cohort.

## 2. Materials and Methods

### 2.1. Study Design

This is a retrospective, multicentric, observational study. We followed up with patients from the start of NAs (lamivudine (LAM), entecavir (ETV), tenofovir disoproxil fumarate (TDF) or tenofovir alafenamide fumarate (TAF)) until the cessation of follow-ups. The primary endpoint was the incidence of HCC.

### 2.2. Patients

A total of 856 patients with HBV chronic infection who were treated with NAs for more than 6 months from 2000 to 2021 were registered from 16 hospitals in Miyagi, Iwate, Akita, Yamagata and Fukushima prefectures in the Tohoku Hepatology Research Meeting (THERME) Study Group [11]. They were consecutive in each hospital, and after the exclusion of patients with hepatitis C virus (HCV) infection or a history of HCC development before the start of NAs and those without sufficient clinical data, a total of 611 patients were analyzed.

### 2.3. Assays of Serological Tests and HBV Genotypes

HBsAg was tested by a chemiluminescent immunoassay (CLIA) with ARCHITECT (Abbott Japan, Tokyo, Japan) or a chemiluminescent enzyme immunoassay (CLEIA) with LUMIPULSE HBsAg-HQ (Fujirebio, Tokyo, Japan). Hepatitis B e antigen (HBeAg) was tested by CLIA with ARCHITECT. HBV DNA levels were quantified by a real-time PCR assay using Cobas TaqMan HBV Auto (Roche Diagnostics, Tokyo, Japan). HBcrAg was tested by CLEIA with LUMIPULSE (Fujirebio). HBV genotypes were determined with an IMMUNIS HBV genotype EIA kit (Institute of Immunology, Tokyo, Japan). As a score for evaluating liver fibrosis, the FIB-4 index was calculated as follows: FIB-4 = age (years) × aspartate aminotransferase (AST, U/L)/[PLT (10^3^/μL) × √alanine aminotransferase (ALT, U/L)] [23]. As a risk score for HCC, the aMAP risk score was calculated as follows: aMAP risk score = [(0.06 × age + 0.89 × sex (Male: 1, Female: 0) + 0.48 × {log_10_ [T-bil (mg/dL) × 17.1] × 0.66 + Alb (g/dL) × −0.85} − 0.01 × PLT) + 7.4]/14.77 × 100 [16].

### 2.4. HCC Surveillance

For HCC surveillance, ultrasonography was basically performed at least twice a year. Additionally, computed tomography or magnetic resonance imaging was performed at least once every year if the patient was considered to have liver cirrhosis. Serum alpha-fetoprotein (AFP) was assayed at every visit at least once every 3 months, and if the AFP level was increased, imaging tests were performed more intensively [11].

### 2.5. Statistical Analysis

Statistical analyses were performed using the chi-squared test for a comparison of the proportions between the two groups and a Wilcoxon rank sum test for a comparison of continuous variables between the two groups. Cumulative incidences were estimated using the Kaplan–Meier method and were compared using a log-rank test. A Cox proportional hazards model was used to identify risk factors, and variables with *p* values less than 0.05 in the univariate analysis were input into the multivariate analysis. Propensity scores were calculated using logistic regression with 5 variables (age, sex, ALT, PLT and HBV DNA), and 1:1 propensity score matching was performed [24]. Differences with *p* values less than 0.05 were considered statistically significant. All statistical analyses were performed using JMP version 16.0 (SAS Institute Inc., Cary, NC, USA). Data were plotted and graphed using GraphPad Prism version 9.3.0 (GraphPad Software Inc., La Jolla, CA, USA).

## 3. Results

### 3.1. Patient Characteristics and Validation of the aMAP Risk Score

Of 611 patients, 390 (63.8%) were male and the median age was 53 (interquartile range, 42–62). Clinical characteristics are shown in Table 1. Among 528 patients whose HBV genotypes were determined, 35.4% and 63.3% were genotypes B and C, respectively. ETV was used as the initial NA in 71.0%. Some of the patients were switched to or supplemented with other NAs (data not shown). Additionally, we evaluated the aMAP risk score (low risk: <50, medium risk: 50–60, high risk: >60) [16]. A total of 48 patients developed HCC during the observation period (median 72 months). When characteristics were compared of patients with HCC and those without, age, sex, the presence of diabetes mellitus (DM), T-Bil, Alb, PLT, AFP, FIB-4 index, aMAP risk and initial NA were significantly different (Table 1). The proportion of patients who were treated with LAM as the initial NA was higher in those with HCC than in those without (31.3% vs. 14.2%). The cumulative incidences of HCC occurrence were 7.4%, 10.6% and 15.0% at 5, 10 and 15 years of NA therapies, respectively (Figure 1a). When the FIB-4 index was analyzed with a cut-off value of 2.5 [11], patients with a higher FIB-4 index showed a significantly higher incidence of HCC (Figure 1b). The aMAP risk score was significantly higher in patients who developed HCC than in those who did not (median 64.2 vs. 54.5, *p* < 0.001) (Figure 1c). The cumulative incidences of HCC at 10 years were 21.4%, 7.6% and 3.5% in the high-risk group, the medium-risk group and the low-risk group, respectively (Figure 1d).

### 3.2. Factors Associated with HCC Incidence

We performed univariate and multivariate analyses of risk factors for the occurrence of HCC. The cut-off values of age, Alb, PLT, AFP and FIB-4 index were determined from the receiver operating characteristic curves for HCC occurrence at 5 years after NAs in our previous study [11]. In the univariate analysis, male gender, age (≥48 years), Alb (<4.0 g/dL), PLT (<15 × 10^4^/μL), AFP (≥5.5 ng/mL) and FIB-4 index (≥2.5) were significantly associated with HCC occurrence (Table 2). Additionally, we evaluated responses to NA therapies using HBV DNA undetectability (including positive but under the detection limit) and ALT normalization 1 year after the start of NAs. For the evaluation, three types of criteria for ALT normalization were used: the WHO criteria (male, ≤30; female, ≤19), the American Association for the Study of Liver Diseases (AASLD) criteria (male, ≤35; female, ≤25) and the Japan Society of Hepatology (JSH) criteria (male and female, ≤30). The response rates for NA therapy based on these criteria and undetectable HBV DNA after 1 to 5 years are shown in Appendix A. Whereas HBV DNA positivity at 1 year was not associated with the occurrence of HCC, abnormal ALT of the WHO criteria and JSH criteria at 1 year was significantly associated with HCC occurrence, and the JSH criteria showed the highest hazard ratio among the three criteria (Table 2). In a multivariate analysis, significant factors in the univariate analysis were included, but the FIB-4 index and aMAP risk score were excluded because these parameters are calculated using other significant factors. As for abnormal ALT at 1 year, only the JSH criteria were included. After the multivariate analysis, male gender, age, PLT, AFP and abnormal ALT in the JSH criteria at 1 year were extracted as significant factors for the occurrence of HCC.

### 3.3. Evaluation of ALT Normalization after NAs

Cumulative incidences of HCC were compared between patients with and without ALT normalization of the three types of criteria at 1 year of treatment (Figure 2a–c). The differences were significant when the WHO criteria and JSH criteria were used, and the *p*-value was lower when the JSH criteria were applied. Then we compared the baseline clinical characteristics between patients with and without ALT normalization in the JSH criteria (Table 3). In patients without ALT normalization at 1 year of treatment, the percentage of males was higher (77.6% vs. 58.5%), PLT was lower (16.2 vs. 18.4 ×10^4^/μL) and HBV DNA was significantly lower (5.4 vs. 5.8 log IU/mL). Creatinine (Cr) was higher in patients without ALT normalization, which is probably associated with the higher frequency of males. Of 156 patients with abnormal ALT at 1 year of therapy, a virological breakthrough at that time was found in six patients (LAM, 5 patients; ETV 1 patient).

We also analyzed ALT normalization at 2 years using the three types of criteria. The tendency of HCC incidence with and without ALT normalization was similar with no significant difference (Figure 2d and Appendix A). Therefore, we considered that the evaluation of ALT normalization at 1 year is more useful than that at 2 years for the prediction of HCC occurrence. There was no difference in HCC incidence between patients with positive HBV DNA and those without at 1 year and 2 years (Appendix A). Additionally, we analyzed the HCC incidence of patients whose ALT was abnormal at 1 year but normalized at 2 years. Of 112 patients whose ALT was abnormal at 1 year, 43 patients achieved ALT normalization at 2 years (Appendix A). When cumulative HCC incidences were compared between the 43 patients and those whose ALT was persistently abnormal until 2 years, no difference was found (Appendix A). Therefore, early ALT normalization might be important for the prediction of HCC occurrence. Furthermore, univariate and multivariate logistic regression analyses were performed to find baseline factors associated with the non-achievement of ALT normalization (Table 4). Then, only male gender and a lower HBV DNA level were extracted as independent factors.

Because LAM is associated with virological breakthroughs leading to an elevation of ALT, we compared HCC incidences excluding patients who were treated with LAM (Figure 2e). The result was similar to that from patients, including LAM-treated patients. Subsequently, propensity score matching for the group with ALT normalization (JSH criteria) at 1 year and without was carried out. After the matching, no baseline factors were significantly different (Appendix A). The cumulative HCC incidence of patients without ALT normalization at 1 year was still higher than those with it (Figure 2f).

### 3.4. Combination of aMAP Risk Score and ALT Normalization

Finally, we analyzed HCC incidence in combination with the aMAP risk score and the achievement/non-achievement of ALT normalization (≤30) at 1 year. To judge the risk simply, we divided patients into three groups: risk-0 group (*n* = 103) with aMAP < 50 and ALT normalization; risk-1 group (*n* = 260) with aMAP ≥ 50 or non-achievement of ALT normalization; risk-2 group (*n* = 106) with aMAP ≥ 50 and non-achievement of ALT normalization. The cumulative incidence of HCC was significantly different among the three groups (*p* < 0.001) (Figure 3). The cumulative incidences of HCC at 10 years were 25.6%, 8.0% and 0% in the risk-2 group, the risk-1 group and the risk-0 group, respectively. The distribution of the aMAP risk score in patients with and without HCC development and ALT normalization at 1 year is shown in Figure 3b. Regardless of ALT normalization, the aMAP risk scores were significantly higher in patients with HCC development, and notably, there was no HCC development in patients with both low-risk aMAP scores and ALT normalization.

## 4. Discussion

In this study, in addition to the international standards of ALT normalization (the WHO and AASLD criteria), the JSH criteria were also evaluated, and it was demonstrated that ALT normalization in the JSH criteria at 1 year after the start of NA therapies was associated with HCC development. In addition, the combination with the aMAP risk score could precisely stratify HCC risk. The finding that there were no patients with HCC development in the risk-0 group with both aMAP < 50 and ALT normalization suggests that the frequency of HCC surveillance with imaging tests such as ultrasonography may be reduced, but further replication studies will be necessary.

The association between HCC development and ALT normalization during NA treatment was recently reported among some cohorts [20,21,22]. However, there are various criteria for the ALT cut-off, and it is unknown which criteria are the most useful for predicting HCC development during NA treatment. The present study shows that the JSH criteria were better than the WHO and AASLD criteria in the Japanese cohort. The difference between these criteria is that the cut-off of ALT in females was lower than that in males in the WHO and AASLD criteria. The WHO guidelines were based on a study using samples from blood donors in Italy who were at low risk for non-alcoholic liver diseases and without HBV or HCV [25]. The JSH criteria, in which the ALT cut-off in females was higher than other criteria, might be useful for the prediction of HCC because the risk for HCC was lower in females. The slightly high ALT in female patients with HBV infection might not affect the development of HCC. However, ethnic differences and the timing of evaluation may affect the appropriate cut-off value of ALT.

Although the reason why an abnormal ALT during NA treatment is associated with HCC development has not yet been determined, the influence of metabolic-associated liver diseases (MAFLD) has been suggested [20,22]. A recent report showed that a high body mass index and high cholesterol were associated with the absence of ALT normalization [22]. Metabolic syndrome was reported to increase the risk of liver fibrosis progression in chronic hepatitis B patients [26,27] and liver-related clinical events and death [28]. Therefore, the presence of MAFLD should be evaluated; however, a limitation of the present study was that data relating to the presence of fatty liver, body mass index and amount of alcohol consumption were not available. However, another recent study showed that regardless of fatty liver, an association between early ALT normalization and lower HCC risk was observed [21]. It was also reported that fatty liver was associated with lower cirrhosis and HCC risk [29] and higher HBsAg seroclearance [30]. Therefore, mechanisms other than MAFLD might be present.

It was reported that mutated large hepatitis B surface (LHBs) proteins accumulate in the endoplasmic reticulum and cause ER stress and mitochondrial dysfunction, which may lead to the development of HCC [31,32]. NAs inhibit reverse transcription but do not suppress the expression of viral proteins. Therefore, if mutations or deletions are present in the preS region of the HBV genome, mutated LHBs that continue to be expressed long after the start of NAs might be a cause of hepatocellular injury, leading to sustained abnormal ALT. In the present study, a delayed response of ALT at 2 years was not associated with a lower risk of HCC (Appendix A). Prolonged treatment might gradually reduce the accumulation of mutated proteins, but a delayed response of ALT indicates that the accumulated amount of LHBs might be high enough to trigger carcinogenesis. This point needs to be clarified in a future study. Inhibiting the accumulation of LHBs in ER might be a therapeutic target for preventing HCC [19].

A recent report showed that the gamma-glutamyl transferase (GGT) level at 6 months after the initiation of NAs was associated with HCC development [33]. The GGT levels reflect pro-oxidant activity and are used as a marker of liver injury. Moreover, GGT was reported to be involved in many diseases such as cardiovascular diseases and cancer [34]. The elevated ALT might be associated with the GGT level, but the data on GGT were not obtained in this cohort. Its association should be analyzed in a future study.

Interestingly, this study showed that lower HBV DNA was independently associated with non-achievement of ALT normalization (Table 4). An inverse association between the presence of fatty liver and HBsAg positivity was reported [35,36] and it may partly explain the result. Consistent with these results, a study using a transgenic mouse model showed that fatty liver reduced HBV replication [37]. Another possible reason is that the retention of mutated LHBs might inhibit the release of HBV particles. An in vitro study showed that an increase in the ratio of preS2 deletion mutant to wild-type LHBs led to a decrease in the release of extracellular HBV DNA [31].

In this study, HBV genotype C was predominant (63%), followed by genotype B (35%). The frequency of genotype B was higher than in other areas of Japan [38] and our previous report showed that patients with genotype B HBV had a lower risk of HCC than those with genotype C in an age and sex-matched analysis [11]. Moreover, patients with genotype B HBV showed a higher probability of HBsAg loss [11], which is a treatment endpoint with a reduced risk of HCC [39]. As for ALT normalization in this study, the frequency of genotype C tended to be higher in patients without ALT normalization (66.2% vs. 59.2%), but not significantly. ALT normalization at 1 year seemed to be useful for HCC prediction in both genotypes C and B, although the difference in the cumulative HCC incidence in genotype B patients was not statistically significant (genotype C, *p* < 0.001; genotype B, *p* = 0.081) because of the low number of HCC patients. Larger numbers of HCC patients with genotype B HBV will be required to clarify this point.

There are limitations in the present study in addition to those mentioned above. First, there is a possibility that HBcrAg might not have been adequately evaluated because the baseline data in many patients were missing. The serum level of HBcrAg is known to be associated with the amount of cccDNA in the liver tissue and has been reported to be a predictive marker for HCC development in several cohorts [40,41,42]. Recently, the usefulness of an ultrasensitive method for HBcrAg, even after HBsAg seroclearance, was reported [43]. Second, the data were obtained only annually, and ALT normalization before 1 year was not evaluated.

In conclusion, the non-achievement of ALT normalization 1 year after the start of NA therapy in patients with chronic HBV infection was associated with HCC development. The combination with the aMAP risk score stratified the HCC risk, and HCC development was rare in patients with both low-risk aMAP and ALT normalization. Further validation studies in other cohorts are required to confirm these results.

## Figures and Tables

**Figure 1 jcm-11-02354-f001:**
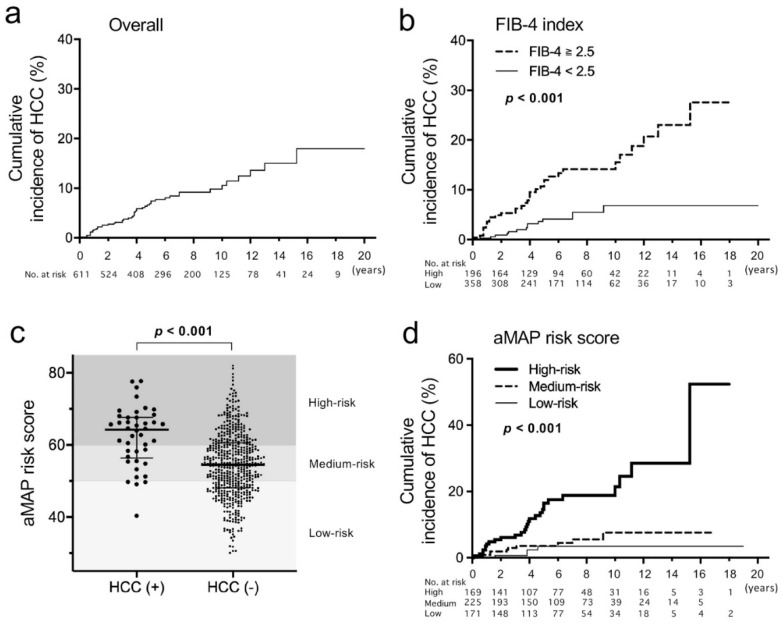
Cumulative incidences of hepatocellular carcinoma (HCC) and age–male–ALBI–platelets (aMAP) risk score in this study. (**a**,**b**) The incidences were estimated using the Kaplan–Meier method in an overall analysis (**a**) and patients were compared with a FIB-4 index ≥2.5 vs. <2.5. (**b**,**c**) Comparison of the aMAP risk score distribution between patients who developed HCC during the observation period and those who did not. Thick lines indicate medians and thin lines indicate interquartile ranges. (**d**) Cumulative incidences of HCC compared among the high-risk group of the aMAP risk score (>60), the medium-risk group (50–60) and the low-risk group (<50). A log-rank test was used to compare cumulative incidences.

**Figure 2 jcm-11-02354-f002:**
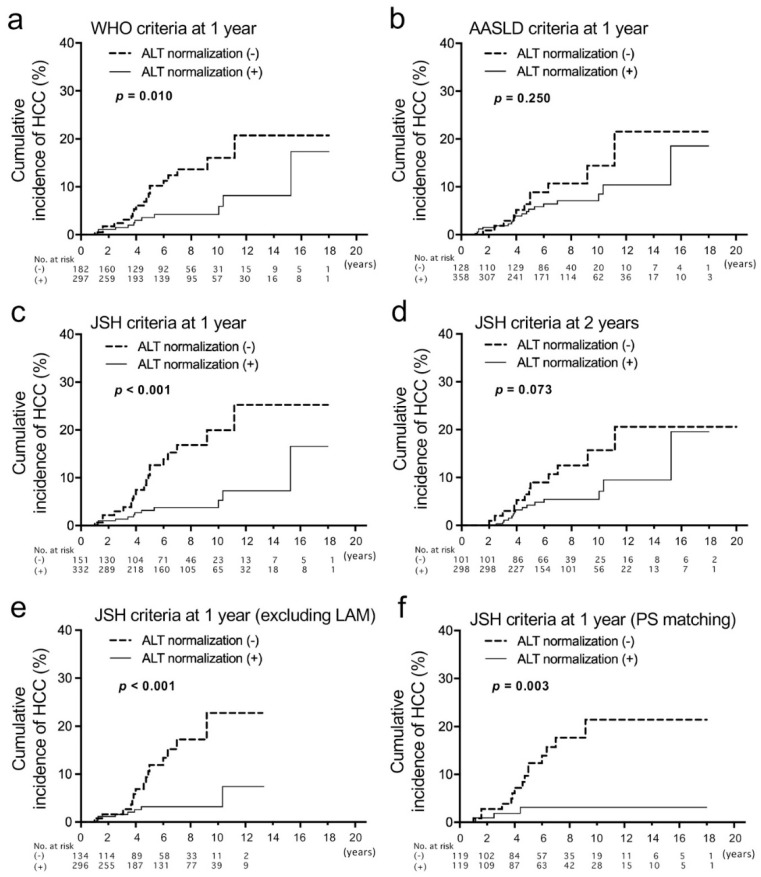
Analyses of cumulative incidences of HCC according to the achievement of alanine aminotransferase (ALT) normalization 1 year after the start of nucleos(t)ide analog therapies that were evaluated with three types of criteria. (**a**–**c**) Comparison of cumulative incidences of HCC between patients who achieved ALT normalization based on the criteria of the World Health Organization (WHO: male, ≤30; female, ≤19) (**a**), American Association for the Study of Liver Diseases (AASLD: male, ≤35; female, ≤25) (**b**) and the Japan Society of Hepatology (JSH: ≤30) (**c**). In these analyses, patients who developed HCC before 12 months and those whose observation periods were less than 12 months were excluded. (**d**) Comparison of cumulative incidences of HCC based on the ALT normalization criteria of JSH at 2 years of therapy. (**e**) Comparison of cumulative incidences of HCC based on the ALT normalization criteria of JSH at 1 year, excluding patients who were treated with lamivudine (LAM). (**f**) Comparison of cumulative incidences of HCC between patients with and without ALT normalization at 1 year based on JSH criteria after propensity score (PS) matching.

**Figure 3 jcm-11-02354-f003:**
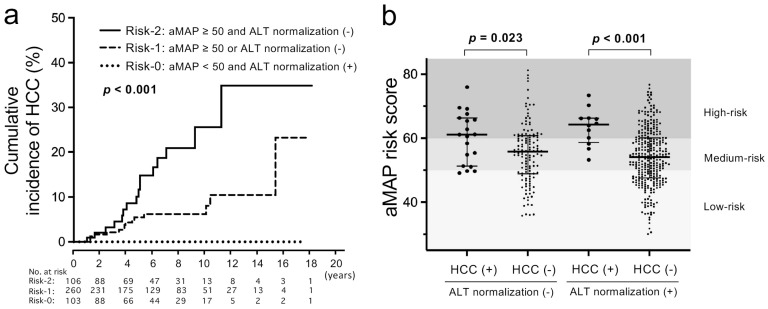
Analyses of cumulative incidences of HCC according to both baseline aMAP risk scores and 1-year ALT normalization of JSH criteria. (**a**) Comparison of cumulative incidences of HCC between the risk-0 group (aMAP risk score < 50 and ALT normalization), the risk-1 group (aMAP risk score ≥ 50 or no ALT normalization) and the risk-2 group (aMAP risk score ≥ 50 and no ALT normalization). (**b**) Distribution of aMAP risk scores in patients with and without ALT normalization at 1 year of NA therapies and with and without HCC development.

**Table 1 jcm-11-02354-t001:** Baseline clinical characteristics of patients with chronic HBV infection at the start of NA therapies.

Parameter	Total (*n* = 611)	HCC (−) (*n* = 563)	HCC (+) (*n* = 48)	*p*-Value *
Age (years)	53 (42–62)	51 (42–61)	58 (52–64)	**0.001**
Sex (male/female)	390/221	349/214	41/7	**<0.001**
Previous IFN (+/−)	84/527	76/487	8/40	0.551
DM (+/−)	35/576	29/534	6/42	**0.035**
T-Bil (mg/dL)	0.8 (0.1–1.1)	0.8 (0.6–1.1)	0.8 (0.7–1.4)	**0.008**
AST (U/L)	49 (34–91)	47 (33–90)	57 (41–131)	0.107
ALT (U/L)	63 (39–129)	64 (39–130)	53 (38–114)	0.764
Alb (g/dL)	4.2 (3.9–4.4)	4.2 (3.9–4.5)	3.9 (3.4–4.2)	**<0.001**
Cr (mg/dL)	0.72 (0.65–0.84)	0.72 (0.64–0.84)	0.78 (0.70–0.86)	0.145
PLT (×10^4^/μL)	17.3 (13.4–21.9)	17.8 (14.0–22.1)	12.3 (9.4–16.2)	**<0.001**
AFP (ng/mL)	4.5 (3.0–9.3)	4.1 (3.0–8.2)	13.5 (6.3–32.6)	**<0.001**
HBV DNA (log IU/mL)	5.7 (4.6–7.0)	5.7 (4.6–7.0)	6.1 (5.1–7.1)	0.368
HBsAg (IU/mL)	940 (162–2875)	840 (148–2910)	1372 (570–1881)	0.542
HBeAg (+/−) †	202/331	191/302	11/29	0.159
HBcrAg (log U/mL)	4.3 (3.0–6.5)	4.4 (3.0–6.6)	3.0 (3.0–5.2)	0.175
FIB-4 index	2.00 (1.25–3.13)	1.89 (1.21–2.94)	3.07 (2.11–5.63)	**<0.001**
aMAP risk (low/medium/high) †	171/225/169	167/215/141	4/10/28	**<0.001**
HBV genotype (A/B/C/D) †	5/187/334/2	5/176/305/2	0/11/29/0	0.477
Initial NA (LAM/ETV/TDF/TAF)	95/434/47/35	80/403/46/34	15/31/1/1	**0.010**
Observation period (months)	72 (36–110)	72 (36–108)	100 (64–141)	**<0.001**

The variables are expressed as mean (interquartile range). AFP, alpha-fetoprotein; Alb, albumin; ALT, alanine aminotransferase; aMAP, age–male–ALBI–platelets; AST, aspartate aminotransferase; Cr, creatinine; DM, diabetes mellitus; ETV, entecavir; HBcrAg, hepatitis B core-related antigen; HBeAg, hepatitis B e antigen; HBsAg, hepatitis B surface antigen; HBV, hepatitis B virus; IFN, interferon; LAM, lamivudine; NA, nucleos(t)ide analog; PLT, platelet counts; TAF, tenofovir alafenamide fumarate; T-Bil, total bilirubin; TDF, tenofovir disoproxil fumarate. * Statistically significant *p*-values are shown in bold type. † The sums of patients are not equal to the total numbers because the data of some of the patients were missing.

**Table 2 jcm-11-02354-t002:** Univariate and multivariate analysis of risk factors for the occurrence of HCC during NA therapies.

Factor	Univariate Analysis		Multivariate Analysis	
	*p*-Value *	HR (95% CI)	*p*-Value *	HR (95% CI)
Male	**0.003**	3.32 (1.49–7.40)	**0.009**	6.91 (1.63–29.23)
Age ≥ 48	**<0.001**	5.04 (1.99–12.87)	**0.014**	3.99 (1.33–11.94)
DM	0.067	2.23 (0.95–5.27)		
Alb < 4.0	**0.001**	2.76 (1.48–5.15)	0.116	1.98 (0.85–4.62)
PLT < 15.0	**<0.001**	4.59 (2.49–8.46)	**0.015**	3.03 (1.24–7.43)
AFP ≥ 5.5	**<0.001**	6.28 (2.42–16.26)	**0.013**	4.40 (1.36–14.20)
HBeAg (+)	0.156	1.65 (0.83–3.31)		
FIB-4 ≥ 2.5	**<0.001**	3.64 (1.91–6.91)	N/A	N/A
aMAP medium or high	**0.007**	4.12 (1.47–11.54)	N/A	N/A
HBV genotype C	0.527	1.25 (0.62–2.52)		
HBV DNA (+) at 1 year	0.824	1.09 (0.51–2.30)		
Abnormal ALT at 1 year				
WHO criteria	**0.007**	2.45 (1.28–4.69)	N/A	N/A
AASLD criteria	0.283	1.44 (0.74–2.82)		
JSH criteria	**<0.001**	3.39 (1.77–6.50)	**0.007**	2.89 (1.33–6.26)

AASLD, American Association for the Study of Liver Diseases; AFP, alpha-fetoprotein; Alb, albumin; ALT, alanine aminotransferase; aMAP, age–male–ALBI–platelets; CI, confidence interval; DM, diabetes mellitus; HBeAg, hepatitis B e antigen; HBV, hepatitis B virus; HR, hazard ratio; JSH, the Japan Society of Hepatology; N/A, not applicable; PLT, platelet counts; WHO, World Health Organization. * Statistically significant *p*-values are shown in bold type.

**Table 3 jcm-11-02354-t003:** Comparison of baseline characteristics between patients who achieved ALT normalization (JSH criteria, ≤30) and those who did not at 1 year of NA therapies.

Parameter	ALT ≤ 30 at 1 Year (*n* = 337)	ALT > 30 at 1 Year (*n* = 156)	*p*-Value *
Age (years)	54 (43–63)	51 (41–61)	0.132
Sex (male/female)	197/140	121/35	**<0.001**
Previous IFN (+/−)	37/300	16/140	0.801
DM (+/−)	14/323	12/144	0.102
T-Bil (mg/dL)	0.8 (0.6–1.0)	0.8 (0.6–1.1)	0.088
AST (U/L)	50 (33–98)	45 (33–70)	0.288
ALT (U/L)	69 (36–140)	57 (42–96)	0.236
Alb (g/dL)	4.2 (3.9–4.4)	4.2 (3.9–4.5)	0.330
Cr (mg/dL)	0.70 (0.60–0.82)	0.79 (0.70–0.87)	**0.002**
PLT (×10^4^/μL)	18.4 (14.5–22.4)	16.2 (12.4–21.8)	**0.023**
AFP (ng/mL)	4.3 (3.0–8.5)	5.5 (3.1–11.1)	0.346
HBV DNA (log IU/mL)	5.8 (4.6–7.1)	5.4 (4.0–6.5)	**0.044**
HBsAg (IU/mL)	978 (198–2851)	928 (145–3262)	0.833
HBeAg (+/−) †	104/182	49/87	0.989
HBcrAg (log U/mL)	4.3 (3.0–6.6)	4.3 (3.0–6.0)	0.564
FIB-4 index	1.97 (1.25–3.21)	2.05 (1.29–2.98)	0.981
aMAP risk (low/medium/high) †	104/131/92	42/60/50	0.499
HBV genotype (A/B/C/D) †	3/115/171/0	2/45/96/2	0.063
Initial NA (LAM/ETV/TDF/TAF)	36/255/23/23	17/116/13/10	0.944
Observation period (months)	63 (36–108)	72 (36–108)	0.332

AFP, alpha-fetoprotein; Alb, albumin; ALT, alanine aminotransferase; aMAP, age–male–ALBI–platelets; AST, aspartate aminotransferase; Cr, creatinine; DM, diabetes mellitus; ETV, entecavir; HBcrAg, hepatitis B core-related antigen; HBeAg, hepatitis B e antigen; HBsAg, hepatitis B surface antigen; HBV, hepatitis B virus; IFN, interferon; LAM, lamivudine; NA, nucleos(t)ide analog; PLT, platelet counts; TAF, tenofovir alafenamide fumarate; T-Bil, total bilirubin; TDF, tenofovir disoproxil fumarate. The variables are expressed as mean (interquartile range). * Statistically significant *p*-values are shown in bold type. † The sums of patients are not equal to the total numbers because the data of some patients were missing.

**Table 4 jcm-11-02354-t004:** Univariate and multivariate logistic regression analysis of baseline factors associated with non-achievement of ALT normalization (JSH criteria) at 1 year.

Factor	Category	Univariate Analysis		Multivariate Analysis	
		*p*-Value *	OR (95% CI)	*p*-Value *	OR (95% CI)
Sex	Male	**<0.001**	2.46 (1.59–3.79)	**<0.001**	2.36 (1.45–3.84)
DM	Presence	0.112	1.92 (0.87–4.26)		
T-bil	By 1.0 mg/dL up	0.440	1.10 (0.87–1.40)		
Cr	By 1.0 mg/dL up	0.487	0.91 (0.67–1.22)		
PLT	By 1.0×10^4^/μL up	**0.048**	0.97 (0.94–1.00)	0.180	0.98 (0.95–1.01)
HBV DNA	By 1.0 log IU/mL up	**0.031**	0.87 (0.78–0.99)	**0.016**	0.86 (0.76–0.97)
HBV genotype	C	0.153	1.35 (0.89–2.05)		

Cr, creatinine; DM, diabetes mellitus; HBV, hepatitis B virus; OR, odds ratio; PLT, platelet counts; T-Bil, total bilirubin. * Statistically significant *p*-values are shown in bold type.

## Data Availability

All data used in this study are available from the corresponding author upon reasonable request.

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
