# Peer review of "Non-Achievement of Alanine Aminotransferase Normalization Associated with the Risk of Hepatocellular Carcinoma during Nucleos(t)ide Analogue Therapies: A Multicenter Retrospective Study"

_jcm, 2022, doi:10.3390/jcm11092354_

Round 1
Reviewer 1 Report
Materials and Methods
2.2. Assay...genotypes.
Why the authors decided to use two different ELISA for HBsAg test? Line 92-94.
Results.
Table 1:
- The authors are requested to use a conventionnal abbreviation for lamuvidine: Not "LAM" but "3TC". Finally, LAM must be replaced by 3TC every where in the paper.
- The authors wrote that the total number of patient is 611, for HCC (-) is 563, and for HCC (+). Could the authors explain why the number of patients changes from one parameters to others, such as "previous IFN (+/-), HBeAg (+/-)?
Table 2:
HBV genotype C is listed as a risk factor for occurence of HCC. Unfortunately, ther is no sentence about this factor in the paragraph 3.2. The authors are requested to notify it in the text.
3.2. Factors...incidence
Why the authors decided to use 3 criteria and not more? Why WHO, AASLD and JSH?
Table 3:
Same remarks concerning the total number and the variation among different parameters, as in the table 1.
Table 5:
This table was not find in the text. Could the authors check it?
Discussion.
A new review titled "HBsAg Loss as a Treatment Endpoint for Chronic HBV Infection: HBV Cure", https://doi.org/10.3390/v14040657 is recently published. The authors are invited to consider this review to support discussion of their results.
Author Response
We thank the Reviewers for the helpful suggestions that have improved this manuscript. We have taken all of these comments to heart through editorial modifications. We hope these are sufficient but can make additional alterations if need be. A point by point response is below. Reviewer comment: Why the authors decided to use two different ELISA for HBsAg test? Line 92-94. Response: Thank you for the comment. A CLIA method had been used mainly before 2014, but a CLEIA method, which is more sensitive than the previous method, became commercially available in 2014 in Japan. However, the timing of introduction was different in each hospital and two different methods were described in the materials and method section in such a way. Basically, the values from these two methods are almost same and shown in the same unit (IU/mL).Reviewer comment: The authors are requested to use a conventionnal abbreviation for lamuvidine: Not "LAM" but "3TC". Finally, LAM must be replaced by 3TC every where in the paper. Response: We appreciate the suggestion. We realize that lamivudine is abbreviated as “3TC” mainly in HIV-associated studies, but “LAM” is also generally used in many HBV-associated studies, at least in Asia (An expert review on the use of tenofovir alafenamide for the treatment of chronic hepatitis B virus infection in Asia, Charlton MR, et al. J Gastroenterol 2020;55:811-823). Therefore, we chose to use “LAM” in this paper. However, if this is not suitable scientifically in this journal, we will be able to replace it.
Reviewer comment: Table 1 -- The authors wrote that the total number of patient is 611, for HCC (-) is 563, and for HCC (+). Could the authors explain why the number of patients changes from one parameters to others, such as "previous IFN (+/-), HBeAg (+/-)? Response: Thank you for the kind comment. We are sorry for the mistake of the numbers regarding “previous IFN”. Now we have corrected them. As for HBeAg, aMAP risk and HBV genotype, data of a part of patients are missing. To understand easily, we added a description as “The sums of patients are not equal to the total numbers because data of some patients were missing” in the footnote of the table.
Reviewer comment: HBV genotype C is listed as a risk factor for occurence of HCC. Unfortunately, ther is no sentence about this factor in the paragraph 3.2. The authors are requested to notify it in the text. Response: In this study, HBV genotype C was not associated with the occurrence of HCC as shown in Table 2. Therefore, we did not mention this point.
Reviewer comment: Why the authors decided to use 3 criteria and not more? Why WHO, AASLD and JSH? Response: We appreciate the important comment. The normal values of ALT are different among institutes, but WHO and AASLD criteria of ALT normal value have been used internationally in many studies in which antiviral responses to NAs were evaluated (Treatment and Renal Outcomes Up to 96 Weeks After Tenofovir Alafenamide Switch From Tenofovir Disoproxil Fumarate in Routine Practice. Toyoda H, et al. Hepatology. 2021;74:656-666). Therefore, we thought that these two are enough in general, but additionally, we analyzed with our standard in Japan (JSH criteria). Then it was revealed that the JSH criteria was the most usable for the prediction of HCC in this Japanese cohort. We stated this point in the discussion as “In this study, in addition to the international standards of ALT normalization (the WHO and AASLD criteria), the JSH criteria was evaluated, and it was demonstrated that ALT normalization in the JSH criteria at 1 year after the start of NA therapies was associated with HCC development”.
Reviewer comment: Table 3 -- Same remarks concerning the total number and the variation among different parameters, as in the table 1. Response: This is due to the reason that is mentioned above. Similarly, we added a description as “The sums of patients are not equal to the total numbers because data of some patients were missing” in the footnote.
Reviewer comment: Table 5 -- This table was not find in the text. Could the authors check it? Response: We are sorry for the mistake. “Table 5” was corrected to “Table 4”.
Reviewer comment: A new review titled "HBsAg Loss as a Treatment Endpoint for Chronic HBV Infection: HBV Cure", https://doi.org/10.3390/v14040657 is recently published. The authors are invited to consider this review to support discussion of their results. Response: Thank you for the kind suggestion. We added a sentence “Also, patients with genotype B HBV showed a higher probability of HBsAg loss, which is a treatment endpoint with a reduced risk of HCC“ in the discussion sectionand added the reference.
Reviewer 2 Report
Non-achievement of alanine aminotransferase normalization is 2 associated with risk of hepatocellular carcinoma during nu-3 cleos(t)ide analogue therapies: a multicenter retrospective 4 study 5
Journal of Clinical Medicine
Thank you for asking me to review the above-titled manuscript. The topic looks interesting. However, there are significant problems in the manuscript.
- Abstract- 1) What is the research question? 2) What were other parameters studied other than normalization of ALR </-30 U/L at 1 year? 3) why ALT was not compared against other parameters such as aMAP?
- English needs editing of the whole manuscript.
- Methods- 1) State study design. 2) State how these 856 patients were selected? 3) Add citations to sections 2.1, 2.3 and 2.4.
- Discussion- 1) What are the limitations of the study?
Author Response
We thank the Reviewers for the helpful suggestions that have improved this manuscript. We have taken all of these comments to heart through editorial modifications. We hope these are sufficient but can make additional alterations if need be. A point by point response is below.Reviewer comment: Abstract- 1) What is the research question? Response: The research question of this study is to find patients under NA therapies who has a high risk of hepatocellular carcinoma. To show this clearly, we modified a sentence in the abstract as “Patients with chronic hepatitis B virus (HBV) infection who are treated with nucleos(t)ide analogues (NAs) still have a risk for hepatocellular carcinoma (HCC) and it is clinically required to distinguish patients with high risk of HCC”.
Reviewer comment: Abstract-2) What were other parameters studied other than normalization of ALR </-30 U/L at 1 year? Response: Because of a word count limit, we have simply stated “The incidences of HCC occurrence were analyzed with clinical factors including on-treatment responses”. It was difficult to include the detailed parameters in the abstract.
Reviewer comment: Abstract-3) why ALT was not compared against other parameters such as aMAP? Response: Thank you for the comment. We added aMAP risk in Tables 1, 2 and 3, and to show that point, we added a sentence “Alanine aminotransferase (ALT) normalization based on the criteria of 3 guidelines was analyzed with other parameters including the age-male-ALBI-platelets (aMAP) risk score” in the abstract.
Reviewer comment: English needs editing of the whole manuscript. Response: According to the comment, we asked a native speaker for the English editing again.
Reviewer comment: Methods-1) State study design. Response: We appreciate the comment. We added a sentence “Patients were followed up since the start of NAs until lost follow-up. The primary endpoint was HCC incidence” in the materials and methods.
Reviewer comment: Methods-2) State how these 856 patients were selected? Response: These patients are consecutive cases in each hospital. We stated as “They are consecutive cases in each hospital and after the exclusion of patients with hepatitis C virus (HCV) infection or a history of HCC development before the start of NAs, and those without sufficient clinical data, a total of 611 patients were analyzed”.
Reviewer comment: Methods-3) Add citations to sections 2.1, 2.3 and 2.4. Response: We added references in these sections.
Reviewer comment: Discussion-1) What are the limitations of the study? Response: We had already described some limitations of this study in the discussion section on page 11 as “First, there is a possibility that HBcrAg might not have been evaluated adequately because the baseline data in many patients were missing. The serum level of HBcrAg is known to be associated with the amount of cccDNA in the liver tissue and has been reported to be a predictive marker for HCC development in several cohorts. Recently, the usefulness of an ultrasensitive method for HBcrAg, even after HBsAg seroclearance, was reported. Second, the data were obtained only annually and ALT normalization before 1 year was not evaluated”.
Round 2
Reviewer 2 Report
Thank you for submitting an amended version of the above-titled manuscript. The authors have addressed most of the issues raised by the reviewers. However, there are still minor issues.
- Still, the English in the manuscript is not adequately addressed. No punctuations in nearly all sentences affect the meaning and the reading of a sentence. Also, a sentence such as "HCC occurs only rarely if HBsAg is cleared." is not grammatically correct. The whole manuscript needs editing.
- The authors DID NOT include a "research question" in the abstract nor the manuscript. What they did is a statement, not a QUESTION.
- A "Study design" was not added as requested; a new item (2.1 Study design) to explain the work's overall design is needed. Change the next items in the methods to 2.2, 2.3 etc.
- Table 2- items that are (N/A), or not addressed. explain why
- The Conclusion, page 10 - "Conclusion, the non-achievement of ALT normalization 1 year after the start of NA 346 therapy was associated with HCC development. The combination with the aMAP risk 347 score stratified the HCC risk and HCC development was rare in patients with both low-348 risk aMAP and ALT normalization. Further validation studies in other cohorts are re-349 quired to confirm these results. ". The authors did not mention in "chronic HBV infection." and left it open. Also, the statement - "The combination with the aMAP risk 347 score stratified the HCC risk" is not clear, and there are grammatical errors—that need editing.
- Please check the discussion for improvement and editing.
Author Response
We thank the Reviewer again for the additional suggestions that have improved this manuscript further. We have taken all of these comments to heart through editorial modifications including English editing. We hope these are sufficient but can make additional alterations if need be. A point by point response is below.
Reviewer comment: Still, the English in the manuscript is not adequately addressed. No punctuations in nearly all sentences affect the meaning and the reading of a sentence. Also, a sentence such as "HCC occurs only rarely if HBsAg is cleared." is not grammatically correct. The whole manuscript needs editing. Response: Thank you for the kind comment. We asked for English editing again through MDPI language editing service. We believe that the English in the manuscript has been edited adequately.
Reviewer comment: The authors DID NOT include a "research question" in the abstract nor the manuscript. What they did is a statement, not a QUESTION. Response: Thank you for the comment. We modified a part of first sentence in the abstract as “it has been clinically questioned whether patients with a high risk of HCC can be identified efficiently“ to show a clinical question clearly. Similarly, such a description was added in the introduction.
Reviewer comment: A "Study design" was not added as requested; a new item (2.1 Study design) to explain the work's overall design is needed. Change the next items in the methods to 2.2, 2.3 etc. Response: Thank you for the suggestion. We added a new item of study design in the materials and methods.
Reviewer comment: Table 2- items that are (N/A), or not addressed. explain why. Response: We appreciate the kind comment. In Table 2, FIB-4 index and aMAP risk score were excluded for the multivariate analysis because these were calculated using other significant factors in the univariate analysis. Also, for abnormal ALT, only JSH criteria was included because this showed the highest hazard ratio. We added sentences in the results as “In a multivariate analysis, significant factors in the univariate analysis were included, but the FIB-4 index and aMAP risk score were excluded because these parameters are calculated using other significant factors. As for abnormal ALT at 1 year, only the JSH criteria were included.”
Reviewer comment: The Conclusion, page 10 - "Conclusion, the non-achievement of ALT normalization 1 year after the start of NA 346 therapy was associated with HCC development. The combination with the aMAP risk 347 score stratified the HCC risk and HCC development was rare in patients with both low-348 risk aMAP and ALT normalization. Further validation studies in other cohorts are re-349 quired to confirm these results. ". The authors did not mention in "chronic HBV infection." and left it open. Also, the statement - "The combination with the aMAP risk 347 score stratified the HCC risk" is not clear, and there are grammatical errors—that need editing. Response: Thank you for these specific comments. We added “in patients with chronic HBV infection” in the conclusion. Also, the sentence "The combination with the aMAP risk score…" was edited to add a comma by the native English speaking editor.
Reviewer comment: Please check the discussion for improvement and editing. Response: We asked for further editing including the discussion and it has been modified.